# Nicotine Dependence and Factors Related to Smoking Cessation among Physicians in Estonia

**DOI:** 10.3390/ijerph17093217

**Published:** 2020-05-06

**Authors:** Mariliis Põld, Kersti Pärna

**Affiliations:** Institute of Family Medicine and Public Health, University of Tartu, Ravila 19, 50411 Tartu, Estonia; kersti.parna@ut.ee

**Keywords:** smoking, physicians, nicotine dependence, Fagerström Test for Nicotine Dependence, factors related to smoking cessation, Estonia

## Abstract

Smoking withdrawal can be difficult due to nicotine dependence (ND). The study objective was to describe ND and to analyze the association between ND and factors related to smoking cessation among daily smoking physicians in Estonia. Data was collected in 2014, using cross-sectional postal survey sampling all practicing physicians (*n* = 5666) in Estonia, of whom 2939 responded (corrected response rate 53.1%). The study sample was restricted to daily smoking physicians (*n* = 171). Results of the Fagerström Test for Nicotine Dependence (FTND) were described and ND scores calculated. Logistic regression was used to determine the association of ND (at-least-moderate vs. low) with factors related to smoking cessation. Crude and fully adjusted ORs with 95% CIs were calculated. The mean FTND score was 2.8 ± 2.1. The odds of having at-least-moderate ND decreased significantly with each year postponing smoking initiation (OR = 0.82, 95% CI 0.72–0.94). After adjustment, ND was no longer associated with the desire to quit smoking and motives to quit. In conclusion, more than half of daily smoking physicians had low ND. Higher ND was associated with younger age of smoking initiation. Knowledge of ND and factors related to smoking cessation is useful in the prevention of smoking and in development of cessation counselling tailored for physicians.

## 1. Introduction

Physicians are considered to have a significant role in tobacco prevention and cessation [1]. However, when physicians are smokers themselves, it is more likely that patients’ smoking habits remain unaddressed [2]. Therefore, it is important that physicians do not smoke. According to the International Classification of Diseases (ICD−10), tobacco smoking as a substance use disorder is considered a mental and behavioral disease (code F17) [3]. One of the components of smoking addiction is nicotine dependence (ND) which often makes quitting difficult [4]. Although the prevalence of smoking has decreased globally [5], in the future, smokers might have higher dependence, as those who have difficulties quitting may remain smokers [4].

Estonia joined the worldwide Convention on Tobacco Control in 2005. Tobacco policy in Estonia can be described as strict, for example, in terms of advertisement bans and picture warnings on cigarette packages. Smoking is forbidden in public institutions like schools, or restricted, for example in hospitals. Smoking cessation services are under further development, and, at the beginning of 2020, a national campaign was carried out to promote quitting smoking. In order for health care specialists to provide the best suitable smoking cessation services and treatment, strength of dependence should be determined. This can be measured, for example, by asking how soon after waking up a person smokes their first cigarette. Time to first cigarette (TTFC) as a measure is considered to have great validity [6]. Quicker time to first cigarette means higher dependence. However, one of the widely used basic six-item dependence measures is the Fagerström Test for Nicotine Dependence (FTND) [7] renamed as the Fagerström Test for Cigarette Dependence in 2012 [8]. As nicotine is one of the main determinants in smoking addiction, ND is addressed in the current paper and therefore the test measure is referred to as FTND. Based on the FTND results, higher ND score indicates lower abstinence rates [9] and reflects more intense withdrawal symptoms [10].

In research on ND, the age of tobacco uptake, the speed and magnitude of nicotine delivery, the development of physical dependence and stimuli-linked associations are among the factors considered [11]. Factors related to smoking cessation, such as the desire to quit, previous quit attempts and smoking relapses, have been found to be associated with smoking dependence [12].

The age-standardized prevalence of daily smoking among physicians in Estonia was 18.4% in 2002 and 11.8% in 2014 among men and 6.2% in 2002 and 4.4% in 2014 among women [13]. Among the general population in Estonia, the age-standardized prevalence of daily smoking was 46.0% in 2002 and 32.9% in 2014 among men and 19.0% in 2002 and 16.5% in 2014 among women [14]. In Europe, the average age-standardized prevalence of daily smoking among 25–64 year-old men was 27.8% and among women, 19.8% in 2014 [15]. Questions of the FTND were included in this survey in 2014, giving a great opportunity to analyze ND among physicians in Estonia. The objective of the current paper is to describe ND and to analyze association between ND and factors related to smoking cessation among daily smoking physicians in Estonia.

## 2. Materials and Methods

### 2.1. Study Design

Study data were drawn from Estonian physicians’ cross-sectional postal smoking survey conducted in 2014. The survey was approved by the Research Ethics Committee of the University of Tartu (Decision No. 235/T–12). Along with the questionnaires, the recipients received an informed consent form including a description of the study design and how the collected data would be used. Respondents were informed that participation in the study would constitute consent. Based on the Estonian Health Care Professionals Registry, all practicing physicians were included in the initial sampling. Questionnaires were mailed to physicians’ (*n* = 5666) home addresses, which were retrieved via data linked to the Estonian population register (Figure 1). In total, 2903 physicians participated in the study. The crude response rate was 51.9%. The corrected response rate (excluding the physicians who were unavailable, had retired, had an incorrect address, had left Estonia or had died) was 53.1%. FTND questions were asked of respondents who answered ‘yes’ to whether they smoked every day (*n* = 171). Double-entering was used to ensure the quality of the data.

### 2.2. Study Variables

The outcome variable was ND measured by the six-item FTND. The test consisted of following questions: ‘How soon after you wake up do you smoke your first cigarette?’; ‘How many cigarettes a day do you smoke on average?’; ‘Do you find it difficult to refrain from smoking in the places where it is forbidden (e.g., on the airplane, in the cinema)?’; ‘Which cigarette would you hate most to give up?’; ‘Do you smoke more frequently during the first hours after waking than during the rest of the day?’; ‘Do you smoke if you are so ill that you are in bed most of the day?’. Based on the answers, an ND score in the range of 0 to 10 was produced. In the present study, the ND was determined to be low if the score was less than 3, moderate if the score was 4–6 and high if the score was 7 or more [9]. For logistic regression analysis, the ND was dichotomized into two categories: low and at-least-moderate (moderate, high).

The following factors were included in the analysis: age of initiation of smoking (continuous variable); desire to quit (yes, no, cannot say); main motives to quit smoking (single-choice question with the following options: personal health problems; wish to set a good example; other reasons (material stimulus, increase in the price of tobacco products, social pressure), and cannot say); number of previous quit attempts (none, 1–2, 3–4, 5 and more); and stress as the main reason to restart smoking (yes, no). Gender (male, female), age (continuous variable), ethnicity (Estonian, non-Estonian) and medical specialty (family physician, specialist doctor, dentist, other) were included as background factors.

### 2.3. Data Analysis

Descriptive analysis was conducted separately for men and women since, according to literature, there are gender differences in smoking and smoking dependence [16]. To compensate for response-bias, post-stratification weights were used based on gender and 5-year age groups of the sample of Estonian physicians [17]. Mean age of respondents and mean age of smoking initiation were calculated, along with standard deviations. Mean FTND score with standard deviation was calculated and ND was determined. Fisher exact test was used to test for differences between men and women in distribution of FTND results, and to test for differences between men and women in distribution of ND. A *t*-test was used to test for differences between the mean age of smoking initiation among respondents with low and at-least-moderate ND.

A multiple logistic regression model was used to test the association of ND (at-least-moderate vs. low) with factors related to smoking cessation. In logistic regression models, data were analyzed together for men and women as there was no significant difference between men and women in ND score. The model used dichotomized ND score (at-least moderate vs. low) as a dependent variable and all factors related to smoking cessation (age of initiation, desire to quit, main motives to quit, number of previous quit attempts, stress as the main reason to restart smoking) as explanatory variables. Background factors (gender, age, ethnicity, medical specialty) were considered as controlling variables in the logistic regression model. Odds ratios (OR) with corresponding 95% confidence intervals (CI) were calculated. In the fully adjusted logistic regression model, OR was adjusted for all factors related to smoking cessation and background factors.

In total, 171 questionnaires were included in the analysis (62 men and 109 women), excluding questionnaires that lacked information concerning smoking status and FTND questions (*n* = 13). Data were analyzed using the statistical package Stata V 14.2 [18]. The study methodology followed the Strengthening the Reporting of Observational Studies in Epidemiology statement guidelines for reporting observational studies [19].

## 3. Results

Characteristics of the sample are provided in Table 1. The mean age was 52.7 ± 14.0 among men and among 55.5 ± 11.7 women. Among daily smoking physicians, 63.7% were women. Among men, 74.2%, and among women, 82.6% were of Estonian ethnicity. More than two thirds of men and almost half of women were specialist doctors.

### 3.1. Results of Fagerström Test for Nicotine Dependence (FTND)

Answers to the five questions in the six-item FTND did not differ (*p* > 0.05) between men and women (Table 2). In total, 11.2% of daily smoking physicians smoked their first cigarette of the day 5 min after waking up, 9.4% smoked more than 20 cigarettes a day on average, 11.7% found it difficult to refrain from smoking in places where smoking is forbidden, 42.6% agreed that giving up the first cigarette in the morning was the most difficult and 21.2% of physicians smoked when they were so ill that they had to stay in bed for most of the day. Gender difference was found in the question concerning preference of smoking more in the morning than during the rest of the day. Among men 27.4% and among women 11.9% (*p* = 0.013) smoked more in the morning than during the rest of the day.

The mean ND score for daily smoking physicians was 2.8 ± 2.1. Among men, the mean ND score was 3.2 ± 2.4, and among women it was 2.7 ± 2.0 (*p* = 0.385). More than half (60.8%) of physicians had low ND (Table 3). Men and women did not differ significantly in terms of the distribution of ND (*p* = 0.107). Physicians with low ND had a mean age of smoking initiation of 21.0 ± 5.1, and physicians with at-least-moderate ND 18.4 ± 3.3 (*p* < 0.001).

### 3.2. Associations between ND Dependence and Factors Related to Smoking Cessation

In the fully adjusted logistic regression model, having at-least-moderate ND was significantly associated with age of smoking initiation (Table 4). When the age of smoking initiation increased by one year, the odds of having at-least-moderate ND was lower (OR = 0.82; 95% CI 0.72–0.94). After adjustment, ND was no longer associated with the desire to quit and motives to quit. No significant association was found between ND and number of previous quit attempts.

## 4. Discussion

The study described results of a Fagerström Test for Nicotine Dependence (FTND) and association of ND with factors related to smoking cessation.

Results of the six-item FTND in the current study showed that among daily smoking physicians in Estonia, about one tenth smoked their first cigarette five minutes after waking up, and more than half smoked 10 or less cigarettes a day on average. Most did not find it difficult to refrain from smoking in places where smoking was forbidden, and about 40% said that they would hate the most to give up the first cigarette in the morning. About one fifth smoked more frequently during the first hours after waking in the morning than during the rest of the day and when they were so ill that had to stay in bed for most of the day. These results refer to rather low ND, with an average value of 2.8 among physicians in Estonia, which means that more than half of them had low ND (score ≤ 3). Surveys among physicians in Germany in 2018 [20], Spain in 2015 [21] and Turkey in 2014 [22] presented similar results—low ND scores were found to be prevalent. Compared to the general population worldwide, Estonian physicians’ mean FTND score was similar to those in Germany and Norway (score 2.8 in 1990s) but lower than those in the USA (4.0 in the 1990s) and China (3.1 in 2013) [23,24,25]. Unfortunately, there is no general population ND data for Estonia to add to the comparison. However, it is known that smoking prevalence among physicians in Estonia in 2014 was about four times lower than among general population [26], making Estonia a mature country in terms of tobacco epidemic [27].

The results of the current study showed that nicotine dependence was significantly associated with the age of smoking initiation. The result appeared in both the crude and fully adjusted logistic regression models. The earlier the initiation of smoking was, the higher the odds of having higher ND. This result was not surprising as it is in accordance with previous findings stating that people who initiate smoking at a younger age are more likely to become dependent [28]. Physicians in Estonia who smoked daily and who had at-least-moderate ND began smoking approximately three years earlier than those with low ND. The difference of mean age of smoking initiation in the groups was statistically significant. Previous results from the smoking survey conducted among Estonian physicians showed that, in 2014, physicians started smoking at an earlier age than in 1982 or 2002. At the same time, an increasing number of non-smokers are entering the profession [26], which creates a situation in which those who smoke have done so beginning at a younger age and thereafter stopped, while others in the profession will not start at all.

In the present study, after adjustment, ND was not significantly associated with the desire to quit or motives to quit. In crude models, compared to physicians who wished to quit smoking, the odds of having at-least-moderate ND were almost three times higher among physicians who did not wish to quit. Similar results were found in surveys conducted in the USA, the UK, Canada and Australia, where a lower dependence was shown to be related to a higher probability of intention to quit [29]. In the current study, compared to physicians who stated personal health problems as the main motive to quit, the odds of having at-least-moderate ND were three times higher among those whose main motives to quit were ‘other reasons’. The group ‘other reasons’ included mainly subgroups of physicians who reported material reasons (increase of the price of tobacco products, material stimulus to quit smoking) and social pressure as the main motives to quit. As the subgroups were very small, the responses were categorized into one group and therefore associations between at-least-moderate ND and specific motives to quit could remain undetected.

The present findings showed no association between ND and the number of quit attempts or stress as a reason for relapse. However, some of these factors have been found to be related to ND in previous studies. For example, previous study data reports on the association between work-related stress and smoking intensity among public-sector employees [30], and between work-related stress and the use of addictive substances among physicians [20]. These findings would allow us to hypothesize that ND could be higher among physicians because the profession is considered to cause higher levels of stress. However, this association was not supported in the current study.

When the results of the current study are interpreted, some contextual factors should also be considered. The tobacco policy in Estonia has been consistent. The Estonian Tobacco Act was enforced in 2005 and is being complemented yearly. This has led to, among other things, smoking being restricted in hospitals. Several Estonian hospitals are smoke-free entirely, and many hospitals actively promote smoking cessation among their staff, and, in some cases, pay for cessation treatment. Recently, the budget impact analysis of smoking cessation interventions in Estonia was published. The report focused mainly on effectiveness, safety and cost-effectiveness of the treatment but also proposed that reimbursement for varenicline, bupropion and NRT should be bound to smoking cessation counselling [31]. This document was long awaited by the policy makers and is hoped to provide some insight into further development of cessation services.

According to the previous results, more than half of currently smoking Estonian physicians expressed a desire to quit [32]. The results of the present study, however, indicate that smokers might still be facing difficulties to do so and therefore would benefit from cessation counselling specifically tailored to physicians’ needs. It is acknowledged that smoking addiction involves several components. Even though difficulties quitting smoking are related to ND, the latter is far from being the only determinant in the addiction. Further analysis is needed to explore psychosocial, contextual and personal aspects of smoking addiction among physicians.

The methodological limitations of the present study are as follows. First, a possible self-representation bias of smokers could contribute to underreporting of their smoking habits [33]. Second, the corrected response rate was just over 53% and there is no information on non-respondents’ smoking habits. The prevalence of smoking might be underestimated since non-respondents might systematically differ from the respondents. To compensate for response bias, data were weighted based on gender and age. Third, because of the small sample size, which is related to a low prevalence of smoking among physicians, significant differences might have remained undetected. Despite these limitations, the smoking survey results for Estonian physicians provide an excellent opportunity to analyze ND and factors related to smoking cessation in a sample representing all Estonian physicians thus providing, for the first time, an overview of ND among daily smoking physicians in Estonia. Moreover, physicians comprise a homogenous group in terms of educational background, which can be considered a major strength in a study exploring health behavior.

## 5. Conclusions

More than half of daily smoking physicians in Estonia had low ND. Higher ND among physicians was associated with a younger age of smoking initiation. Knowledge of ND and factors related to smoking cessation is useful in the development of smoking cessation services targeted towards physicians. The present study also contributes to evidence supporting the development of strategies to postpone smoking initiation.

## Figures and Tables

**Figure 1 ijerph-17-03217-f001:**
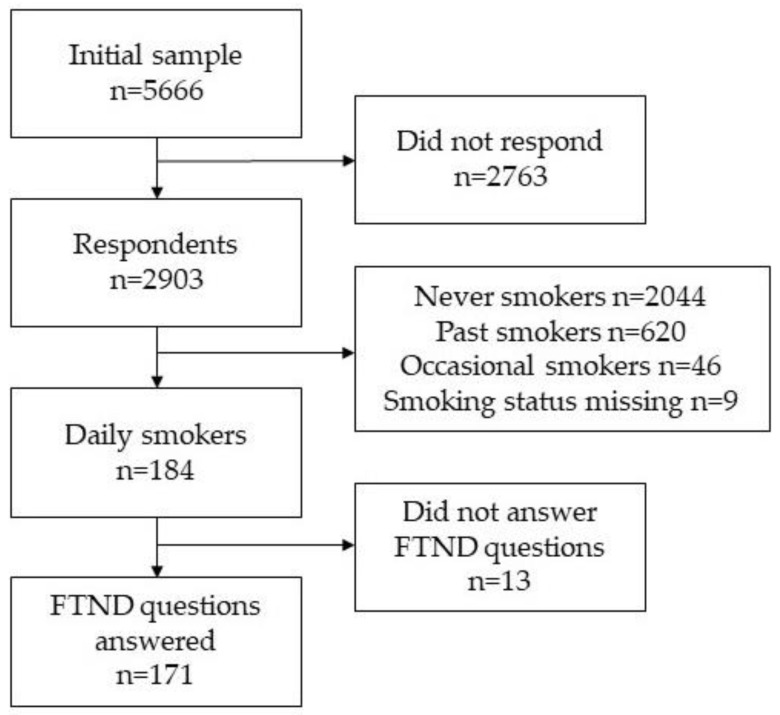
Flow chart of selection of daily smoking physicians from the study sample, postal cross-sectional survey among physicians in Estonia, 2014.

**Table 1 ijerph-17-03217-t001:** Characteristics of the sample.

Characteristic	Men (*n* = 62) %	Women (*n* = 109) %	Total (*n* = 171) %
**Mean Age (cont.), SD**	52.7 ± 14.0	55.5 ± 11.7	54.5 ± 12.6
**Ethnicity**			
Estonian	74.2	82.6	79.5
Non-Estonian	25.8	17.4	20.5
**Medical Specialty**			
Family Physician	9.7	22.9	18.1
Specialist Doctor	66.1	48.6	55.0
Dentist	16.1	21.1	19.3
Other	8.1	7.3	7.6

**Table 2 ijerph-17-03217-t002:** Distribution (%) of the results of the six-item Fagerström Test for Nicotine Dependence (FTND) with points for calculation of nicotine dependence (ND) score among Estonian physicians, 2014.

FTND Items	FTND Points Per Answer	Men %	Women %	*p*-Value	Total %
**How Soon after You Wake Up do You Smoke Your First Cigarette?**					
In 5 min	3	15.3	8.0	0.249	11.2
In 6–30 min	2	29.8	39.1		35.1
In 31–60 min	1	25.7	32.9		29.8
Later	0	29.1	20.0		24.0
**How many Cigarettes a Day do You Smoke on Average?**					
≤10	0	42.2	57.8	0.184	51.1
11–20	1	46.0	34.7		39.6
21–30	2	10.1	6.5		8.1
≥31	3	1.8	1.0		1.3
**Do You Find it Difficult to Refrain from Smoking in the Places Where it is Forbidden (E.G. On The Airplane, in the Cinema)?**					
Yes	1	13.6	10.2	0.617	11.7
No	0	86.4	89.8		88.3
**Which Cigarette would You Hate Most to Give Up?**					
First in the Morning	1	44.8	40.9	0.872	42.6
Any Other	0	55.2	59.1		57.4
**Do You Smoke more Frequently During the First Hours after Waking Than During the Rest of the Day?**					
Yes	1	28.0	11.6	0.013	18.7
No	0	72.0	88.4		81.3
**Do You Smoke if You are so Ill that You are in Bed most of the Day?**					
Yes	1	24.8	18.4	0.552	21.2
No	0	75.2	81.6		78.8

**Table 3 ijerph-17-03217-t003:** Distribution (%) of nicotine dependence (ND) among daily smoking physicians in Estonia, 2014.

ND (Score)	Men	Women	*p*-Value	Total
Low (0–3)	51.6	67.8	0.107	60.8
Moderate (4–6)	41.3	26.8		33.1
High (7–10)	7.1	5.4		6.1

**Table 4 ijerph-17-03217-t004:** Results of logistic regression analysing factors associated with nicotine dependence (ND) (at-least-moderate vs. low) among Estonian physicians smoking daily, 2014.

Variables	ND (%)		Crude OR (95% CI)	Adjusted OR ^a^ (95% CI)
	Low	At-Least-Moderate		
**Factors Related to Smoking Cessation**				
**Age of Smoking Initiation ^b^**	-	-	**0.83 (0.74–0.92)**	**0.82 (0.72–0.94)**
**Desire to Quit**				
Yes	61.8	48.4	1	1
No	15.8	35.5	**2.88 (1.33–6.21)**	1.99 (0.76–5.21)
Cannot Say	22.4	16.1	0.92 (0.40–2.13)	0.85 (0.31–2.35)
**Motives to Quit**				
Personal Health Problems	68.5	66.3	1	1
Wish to Set a good Example	10.2	5.2	0.53 (0.16–1.82)	0.77 (0.19–3.19)
Other	6.2	17.9	**3.00 (1.06–8.52)**	2.48 (0.59–10.39)
Cannot Say	15.1	10.6	0.72 (0.28–1.85)	0.53 (0.19–1.49)
**Previous Quit Attempts**				
None	23.6	34.0	1	1
1–2	43.9	38.9	0.61 (0.29–1.31)	0.56 (0.22–1.40)
3–4	20.6	21.7	0.73 (0.29–1.82)	0.70 (0.23–2.07)
5 and more	11.9	5.4	0.32 (0.07–1.30)	0.27 (0.05–1.44)
**Stress as Main Reason for Relapse**				
No	63.4	64.0	1	1
Yes	36.6	36.0	0.97 (0.50–1.89)	1.66 (0.73–3.77)

^a^ Adjusted for all other variables in the table + background variables (age, gender, ethnicity, medical specialty). ^b^ Continuous variable. Data in bold shows significant findings.

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
