# Peer review of "Nicotine Dependence and Factors Related to Smoking Cessation among Physicians in Estonia"

_ijerph, 2020, doi:10.3390/ijerph17093217_

Round 1
Reviewer 1 Report
The 12-word title of your study is clear and concise. The abstract is concise and presents the study findings and conclusions clearly.
The Introduction provides a relevant context for this manuscript, cited relevant articles and clarifies FTND use clearly.
The Research Design describes a mailed survey, conducted in 2014 thus clarifying where the data were drawn. A survey is appropriate and it may have included questions similar to other previous Estonian surveys. The limitations identified are consistent with survey research.
Materials and Methods. Although this manuscript is based on a subset of questions from a larger postal survey, there must be mention of ethical or organizational clearance from an ethical body or the Health Care Professional Registry. These questionnaires were mailed to thousands of individual physician home addresses. Was subject consent assumed upon return of the completed questionnaire?
A strength of this study was the clear description of the study variables (dependent, explanatory and background) and how they were measured. Were reliability or validity stats run on the data? If so it would be appropriate to present this data. The reporting of descriptive analysis before logistic regression was used is appropriate.
Results. The data in Tables 1, 2 and 3 were clearly presented.
Discussion. On page 6 of 9, lines 58 to 61. I had to read this sentence several times and I still do not find it clear. Please try to restate this sentence.
In Line 68 please replace “proven” with “supported”.
Minor grammar and reference comments.
Line 45. Add “this” in front of “current”
Line 53, 4th word is plural possessive
Line 119, add a comma after “forbidden”
References page 8, Line 126 and 128, please check spelling of journal abbreviation; Line 139, add italics to journal name; line 174, is should there be a period after doi?
Reviewer 2 Report
I am grateful for the opportunity to review the manuscript presented to me. I hope that the comments would be helpful in to publish the manuscript. I believe the paper is worth considering for publication, however requires major revision (comments are indicated in the attached text).

Round 2
Reviewer 2 Report
Article prepared in accordance with the guidelines
